# Single-nucleotide conservation state annotation of the SARS-CoV-2 genome

Soo Bin Kwon [ID] [1,2] & Jason Ernst [ID] [1,2,3,4,5,6,7 ✉]

Given the global impact and severity of COVID-19, there is a pressing need for a better understanding of the SARS-CoV-2 genome and mutations. Multi-strain sequence alignments of coronaviruses (CoV) provide important information for interpreting the genome and its variation. We apply a comparative genomics method, ConsHMM, to the multi-strain alignments of CoV to annotate every base of the SARS-CoV-2 genome with conservation states based on sequence alignment patterns among CoV. The learned conservation states show distinct enrichment patterns for genes, protein domains, and other regions of interest. Certain states are strongly enriched or depleted of SARS-CoV-2 mutations, which can be used to predict potentially consequential mutations. We expect the conservation states to be a resource for interpreting the SARS-CoV-2 genome and mutations.

[1] Bioinformatics Interdepartmental Program, University of California, Los Angeles, CA, USA. [2] Department of Biological Chemistry, University of California, Los Angeles, CA, USA. [3] Eli and Edythe Broad Center of Regenerative Medicine and Stem Cell Research at University of California, Los Angeles, CA, USA. [4] Computer Science Department, University of California, Los Angeles, CA, USA. [5] Department of Computational Medicine, University of California, Los Angeles, CA, USA. [6] Jonsson Comprehensive Cancer Center, University of California, Los Angeles, CA, USA. [7] Molecular Biology Institute, University of California, Los Angeles, CA, USA. ✉email: jason.ernst@ucla.edu

With the urgent need to better understand the genome and mutations of SARS-CoV-2, multi-strain sequence alignments of coronaviruses (CoV) have become available[1] where multiple sequences of CoV are aligned against the SARS-CoV-2 reference genome. Sequence alignments provide important information on the evolutionary history of different genomic bases. Such information can be useful in interpreting mutations, as for example bases with strong sequence constraint or accelerated evolution have been shown to be enriched for phenotype-associated variants[2,3]. While existing systematic annotations that quantify sequence constraint from alignments[4,5] are informative, they reduce the information in the underlying alignment to a single univariate or binary value and thus are limited in the information they convey. Additional information about patterns of which sequences align to and match the SARS-CoV-2 genome at each base may be useful in analyzing the SARS-CoV-2 genome and mutations.

As a complementary approach to sequence constraint scoring methods, ConsHMM was recently introduced to systematically annotate a given genome with conservation states that capture combinatorial and spatial patterns in a multi-species sequence alignment[6]. ConsHMM specifically models whether bases from non-reference sequences align to and match each base in the reference genome. ConsHMM extends ChromHMM, a widely used method that uses a multivariate hidden Markov model (HMM) to learn patterns in epigenomic data de novo and annotate genomes based on the learned patterns[7]. Apart from the input alignments which were generated using phylogenetic trees, ConsHMM does not explicitly use any phylogenetic information and therefore does not make any strict assumptions on the phylogenetic relationship among sequences. This allows ConsHMM to be more flexible in capturing various patterns within alignments than the more commonly used comparative genomics approaches that define a single constraint score or binary calls of constrained elements based on phylogenetic modeling. Previous work applying ConsHMM to multi-species alignment of other genomes have shown that the conservation states learned by ConsHMM capture various patterns in the alignment overlooked by previous methods and are useful for interpreting DNA elements and phenotype-associated variants[6,8].

Motivated by the current need to better understand the SARS-CoV-2 genome and mutations, here we apply ConsHMM to two multi-strain sequence alignments of CoV that were recently made available[1] and learn two sets of conservation states. The first alignment consists of Sarbecoviruses, a subgenus under genus Betacoronavirus in the family of Coronavirdae[9]. This alignment consists of SARS-CoV and other Sarbecoviruses that infect bats aligned to the SARS-CoV-2 genome. The second alignment consists of CoV that infect various vertebrates (e.g. human, bat, pangolin, mouse, birds) aligned to the SARS-CoV-2 genome.

Given the two sets of conservation states learned by ConsHMM from these two alignments, we annotate the SARS-CoV-2 genome with the states and analyze the states' relationship to external annotations to understand their properties. We observe that the states capture distinct patterns in the input alignment data. Using external annotations of genes, regions of interest, and mutations observed among SARS-CoV-2 sequences, we observe that the states also have distinct enrichment patterns for various annotated regions. We generate genome-wide tracks that score each nucleotide based on state depletions and enrichments for observed mutations, which can be used to prioritize bases where mutations are more likely to be consequential. Overall, our analysis suggests that the ConsHMM conservation states highlight genomic bases with distinct evolutionary patterns in the input sequence alignments and potential biological significance. The ConsHMM conservation state annotations and

tracks of state depletion of mutations are resources for interpreting the SARS-CoV-2 genome and mutations.

## Results

**Annotating SARS-CoV-2 with conservation states learned from the alignment of Sarbecoviruses.** First, we annotated the SARS-CoV-2 genome with 30 conservation states learned from a Sarbecovirus sequence alignment, labeled as states S1 to S30 (Figs. 1 and 2; Supplementary Table 1; "Methods"). The Sarbecovirus alignment consists of SARS-CoV and 42 other Sarbecoviruses that infect bats aligned to the SARS-CoV-2 genome (Fig. 2c). The states capture distinct patterns of which Sarbecovirus strains align to and match the SARS-CoV-2 genome (Fig. 2a) and show notable enrichment patterns for external annotations of genes, proteins, and regions of interest within them (Fig. 2b and Supplementary Fig. 1). State S17 corresponds to bases where all strains align to and match SARS-CoV-2 with high probability and appears in the genome most frequently, covering 48% of the genome. Similarly, state S18 annotates bases with high align and match probabilities except it has slightly reduced probability of matching two strains that are most distal from SARS-CoV-2 (SARS-related CoV strain BtKY72 and Bat CoV BM48-31/BGR/2008). Unlike state S17, state S18 is strongly enriched for a region in RNA-dependent RNA polymerase (RdRp) that is known to interact with the antiviral drug remdesivir (tenfold; $P < 0.0001$). State S6 annotates bases where all strains align to SARS-CoV-2 with high probability but only the strain closest to SARS-CoV-2, bat CoV RaTG13, matches SARS-CoV-2 with high probability, highlighting bases with alleles unique to SARS-CoV-2 and bat CoV RaTG13 with respect to other Sarbecoviruses. As expected, state S6 is enriched for the third codon position (2.2-fold; $P < 0.0001$) where derived alleles are less likely to alter the amino acid. In contrast to state S6, state S28 corresponds to bases where bat CoV RaTG13 both aligns to and matches SARS-CoV-2 with high probability but has a low probability of aligning to other Sarbecoviruses. State S28 covers 1% of the genome and highlights bases unique to SARS-CoV-2 and bat CoV RaTG13 with respect to other Sarbecoviruses. Notably, state S28 is highly enriched for human ACE2 binding domain (22-fold; $P < 0.0001$), which is consistent with recent work suggesting that this binding domain is under strong positive selective pressure due to its critical role in host infection[10,11]. State S28 also annotates a region, known as the PRRA motif, that may have been inserted into the SARS-CoV-2 genome, potentially resulting in increased infectiousness[12,13]. We note that state S28 also annotates the first five and the last seventeen bases of the genome, which may reflect technical issues with sequencing the genome ends in some strains[14]. A different state, state S13, corresponds to bases where all strains align to the reference with high probability, but only a specific subset of the strains have the same nucleotide as SARS-CoV-2 with high probability (Fig. 2a). This subset of strains includes Sarbecoviruses that are relatively distal to SARS-CoV-2 while excluding strains that are closer to SARS-CoV-2, corresponding to a deviation along a specific branch of the phylogenetic tree (Supplementary Fig. 2). State S29 shows strong enrichment of intergenic bases (36-fold; $P < 0.0001$) and gene *ORF10* (59-fold; $P < 0.0001$), which is consistent with recent work suggesting that *ORF10* may not be a protein-coding gene based on gene expression[15] and phylogenetic codon modeling[9].

**Annotating SARS-CoV-2 with conservation states learned from the alignment of Coronaviruses infecting vertebrates.** In addition to the 30-state model learned from the Sarbecovirus sequence alignment, we learned another 30-state model by applying ConsHMM to the alignment of 56 CoV from vertebrate

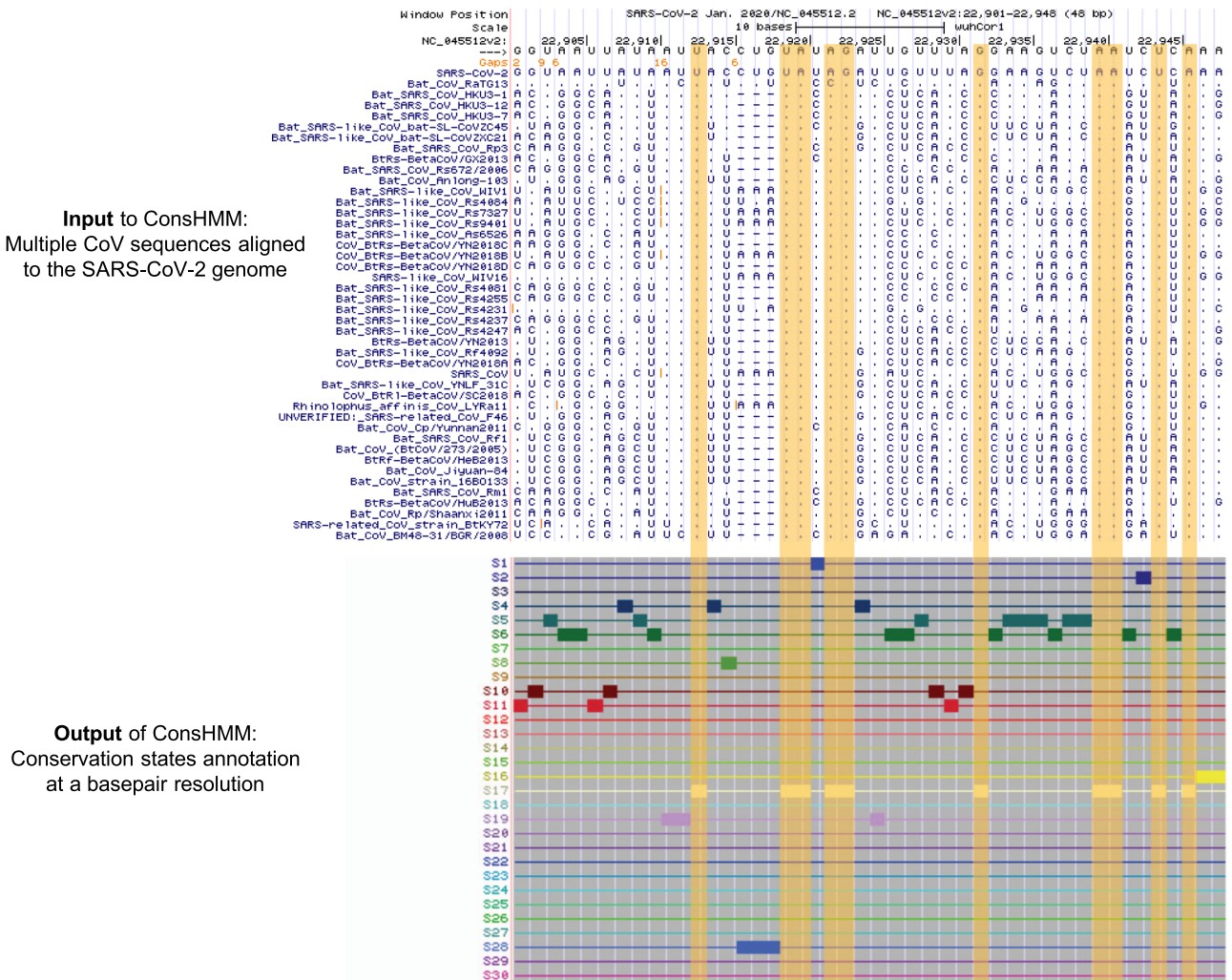

**Fig. 1 Genome browser view of ConsHMM input and output for a portion of the SARS-CoV-2 genome.** Shown is an example portion of the Sarbecovirus sequence alignment input to ConsHMM and ConsHMM's conservation state annotation of the SARS-CoV-2 genome as viewed in the UCSC Genome Browser[1]. The top row of the alignment shows the reference sequence, the SARS-CoV-2 genome. This is followed by 43 rows corresponding to different Sarbecovirus sequences aligned against the reference, representing the 44-way Sarbecovirus sequence alignment. In each of these rows, a horizontal dash is shown at a position if the row's sequence has no base that aligns to the reference base at the position shown in the top row. A dot is shown if the sequence has the same nucleotide as the reference. A specific letter is shown if for that particular base the row's sequence has a different nucleotide than the reference. Below the alignment are 30 ConsHMM conservation states learned from the alignment. Each row corresponds to a state. To demonstrate how bases with similar alignment patterns in the input data are annotated with the same state, bases annotated with state S17 are highlighted in yellow boxes, which have most Sarbecoviruses aligning to and matching the reference with high probabilities.

hosts against SARS-CoV-2 (states V1 to V30; Fig. 3 and Supplementary Table 2; "Methods"). The vertebrate CoV alignment consisted of a diverse set of CoV that included not only Sarbecoviruses, but also CoV that are evolutionarily more diverged from SARS-CoV-2 than Sarbecoviruses (Fig. 3c). We therefore applied ConsHMM separately to the vertebrate CoV alignment, instead of combining the two alignments.

The resulting conservation states correspond to bases with distinct probabilities of various strains of vertebrate CoV aligning to and matching SARS-CoV-2 and exhibit notable enrichment patterns for previously annotated regions within genes (Fig. 3a and Supplementary Fig. 1). State V27 annotates bases in which all 56 CoV align to and match SARS-CoV-2, with a genome coverage of 8%. State V19 corresponds to bases in which specifically the four strains most closely related to SARS-CoV-2 based on phylogenetic distance, which include two bat CoV (RaTG13 and BM48-31/BGR/2008), pangolin CoV, and SARS-CoV, align to and match SARS-CoV-2 with high probabilities. State V20 has

both high align and match probabilities for bat CoV RaTG13 and pangolin CoV and is enriched for the spike protein's receptor-binding domain (RBD), where a recombination event between a bat CoV and a pangolin CoV might have occurred[12] (6.9-fold enrichment; $P < 0.0001$). Additionally, state V29 with high align and match probabilities specifically for bat CoV RaTG13 annotates the PRRA motif mentioned in the previous section, which is consistent with the possibility that the motif was recently introduced to the SARS-CoV-2 genome.

Since the input vertebrate CoV alignment includes several CoV infecting human, the states learned from this alignment can be used to investigate the varying pathogenicity among human CoV. State V14 corresponds to bases shared among pathogenic human CoV, including SARS-CoV-2, SARS-CoV, and Middle East respiratory syndrome-related CoV (MERS-CoV), but not shared among less pathogenic human CoV which are associated with common cold (OC43, HKU1, 229E, and NL63). Bases annotated by this state are candidates for contributing to the shared

**a** Emission parameters learned by 30-state ConsHMM model based on **Sarbecovirus** sequence alignment

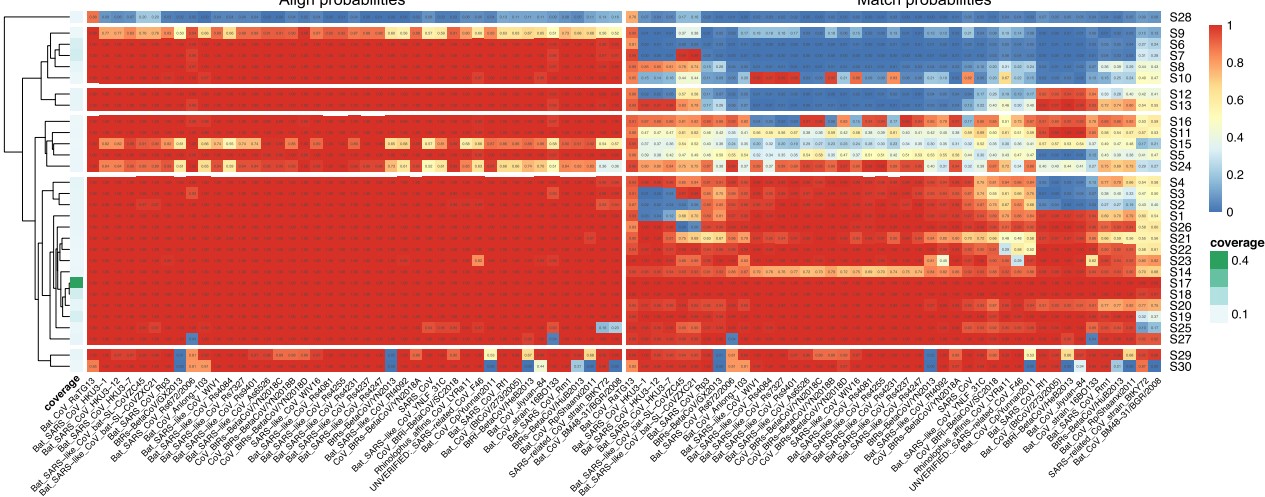

**b** State enrichment for external annotations

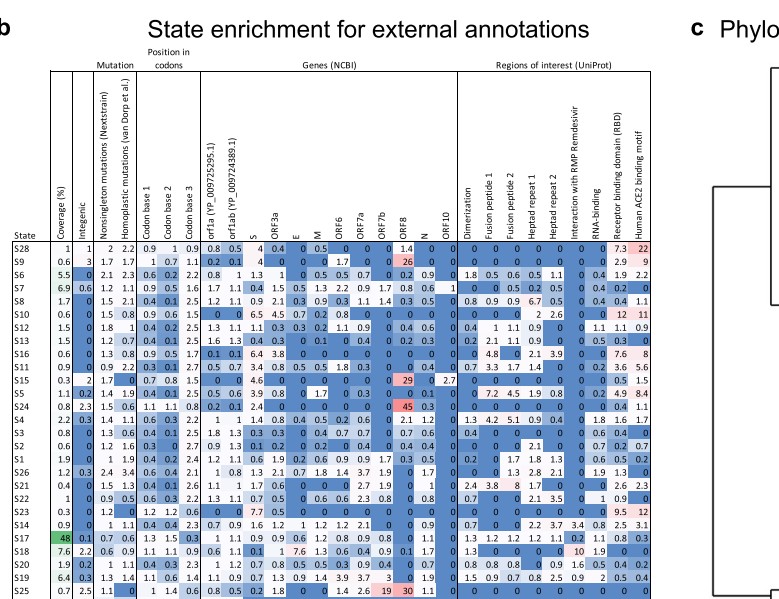

**c** Phylogenetic tree of the aligned Sarbecoviruses

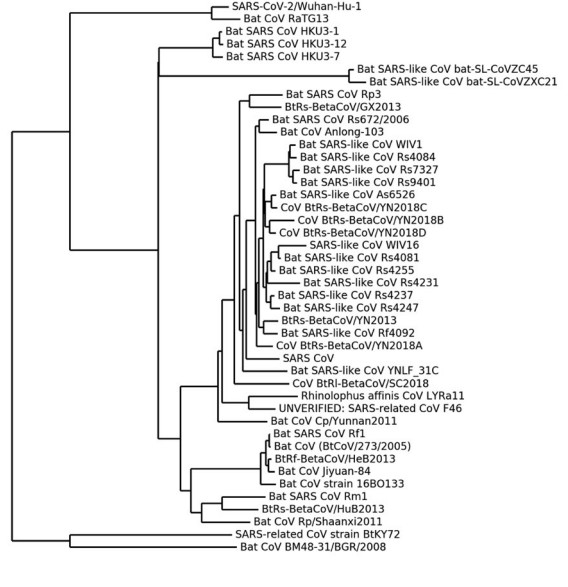

**Fig. 2 ConsHMM conservation states learned from the Sarbecovirus alignment. a** State emission parameters learned by ConsHMM. The left half of the heatmap shows for each state the probability of each CoV strain having a base aligning to a base in the reference, which is SARS-CoV-2. The right half shows for each state the probability of each CoV strain having a base aligning to and matching (having the same nucleotide) a base in the reference. In both halves, each row in the heatmap corresponds to a ConsHMM conservation state with its number on the right side of the heatmap. Rows are ordered based on hierarchical clustering and optimal leaf ordering[34]. In both halves, each column corresponds to SARS-CoV or one of the 42 CoV that infect bats. Columns are ordered based on each strain's phylogenetic divergence from SARS-CoV-2 according to the phylogenetic tree shown in **c**, with closer strains on the left. The column on the left shows the genome-wide coverage of each state colored according to a legend labeled "coverage" on the right. **b** State enrichment for external annotations of mutations, codons, genes, and regions of interest. The first column of the heatmap corresponds to each state's genome coverage, and the remaining columns correspond to fold enrichments of conservation states for external annotations of intergenic regions, mutations, position within codons, NCBI gene annotations[31], and UniProt regions of interest[19]. Each row, except the last row, corresponds to a conservation state, ordered based on the ordering shown in **a**. The last row shows the genome coverage of each external annotation. Each cell corresponding to an enrichment value is colored based on its value with blue as 0 (annotation not overlapping the state), white as 1 to denote no enrichment (fold enrichment of 1), and red as the global maximum enrichment value. Each cell corresponding to a genome coverage percentage value is colored based on its value with white as the minimum and green as the maximum. All annotations were accessed through the UCSC Genome Browser[1] except for nonsingleton mutations from Nextstrain[27] and homoplastic mutations from a prior study[18]. **c** Phylogenetic tree of the Sarbecoviruses included in the alignment. Each leaf corresponds to a Sarbecovirus strain included in the 44-way Sarbecovirus alignment. This tree was obtained from the UCSC Genome Browser[1] and plotted using Biopython[35]. SARS-CoV-2/Wuhan-Hu-1, the reference genome of the alignment, is at the top.

**a** Emission parameters learned by 30-state ConsHMM model based on **vertebrate CoV** sequence alignment

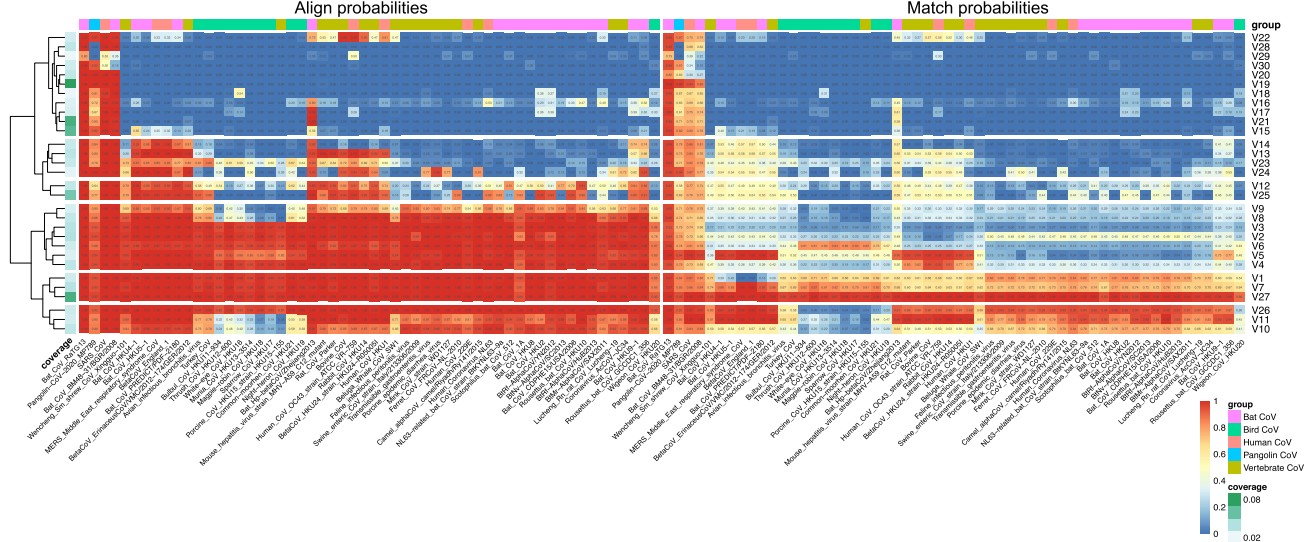

**b** State enrichment for external annotations

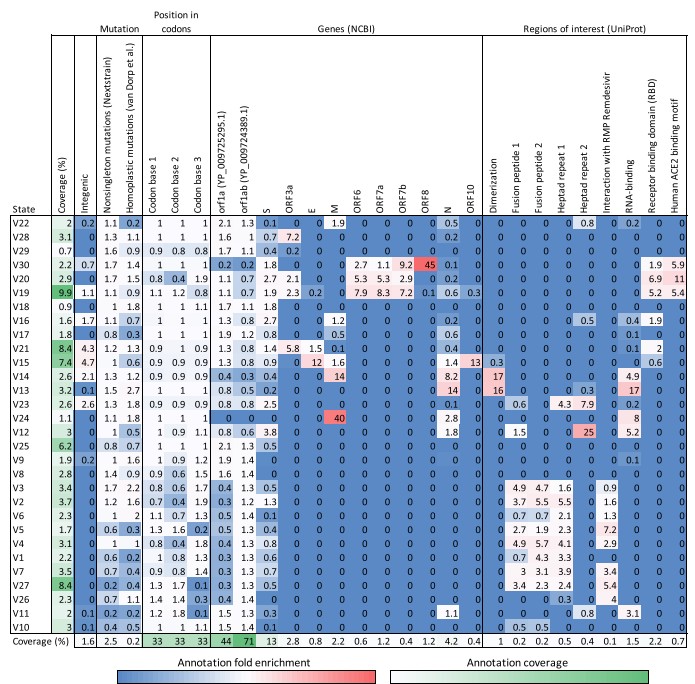

**c** Phylogenetic tree of the aligned vertebrate CoV

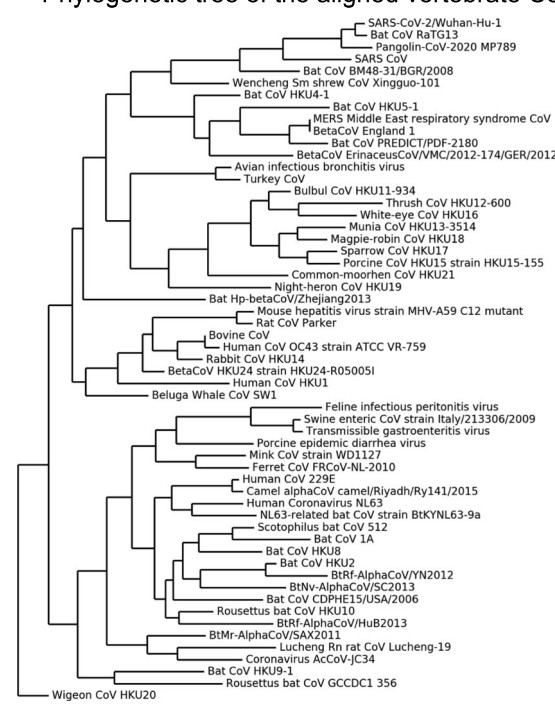

pathogenicity of SARS-CoV, SARS-CoV-2, and MERS-CoV (Supplementary Table 3). We compared bases annotated by this state to positions identified in a previous study that located indels differentiating pathogenic CoV from common-cold-associated CoV using an alignment of 944 human CoV sequences under a supervised learning framework[16]. State V14 overlapped with two insertions identified in that study, one of which is in the nucleocapsid protein and was suggested to contribute to the virus's pathogenicity by enhancing its nuclear localization signals[16] (overlapping positions: 29116–29124). Moreover, using state V14 we identify additional loci potentially unique to pathogenic CoV that were not reported in the previous study (Supplementary Table 3). While this could be explained mostly by the different sequences included in the alignments used here and in the previous study, we find among the additional loci those that are shared among all pathogenic sequences, but missing in all

common-cold-associated sequences according to the previous study's human CoV alignment (Supplementary Table 3; "Methods"). Among such additional loci that are unique to pathogenic sequences, but not previously reported, is an 8-bp region (positions 28416–28423) in the nucleocapsid protein. This protein was shown to enrich for indels specific to pathogenic CoV in the previous study. Overall, this demonstrates the conservation state annotations learned using an unsupervised approach identified additional genomic bases that may contribute to the pathogenicity of CoV infecting humans.

**Conservation states' relationship to standard sequence constraint annotations**. To establish that conservation states contain additional information relative to standard sequence constraint scores, we compared to constraint scores generated by PhastCons[4] and PhyloP[5] and binary constrained elements called

**Fig. 3 ConsHMM conservation states learned from the vertebrate CoV alignment. a** State emission parameters learned by ConsHMM. The left half of the heatmap shows for each state the probability of each CoV strain having a base aligning to a base in the reference, which is SARS-CoV-2. The right half shows for each state the probability of each CoV strain having a base aligning to and matching (having the same nucleotide) a base in the reference. In both halves, each row in the heatmap corresponds to a ConsHMM conservation state with its number on the right side of the heatmap. Rows are ordered based on hierarchical clustering and optimal leaf ordering[34]. In both halves, each column corresponds to one of the 56 CoV that infect vertebrates, excluding SARS-CoV-2. Columns are ordered based on each strain's phylogenetic divergence from SARS-CoV-2 according to the phylogenetic tree shown in **c**, with closer strains on the left. Cells in the top row above the heatmap are colored according to the color legend on the bottom right to highlight specific groups of CoV with common vertebrate hosts. The column on the left shows the genome-wide coverage of each state colored according to a legend in the bottom right. **b** State enrichment for external annotations of mutations, codons, genes, and regions of interest. The first column of the heatmap corresponds to each state's genome coverage, and the remaining columns correspond to fold enrichments of conservation states for external annotations of intergenic regions, mutations, position within codons, NCBI gene annotations[31], and UniProt regions of interest[19]. Each row, except the last row, corresponds to a conservation state, ordered based on the ordering shown in **a**. The last row shows the genome coverage of each external annotation. Each cell corresponding to an enrichment value is colored based on its value with blue as 0 (annotation not overlapping the state), white as 1 to denote no enrichment (fold enrichment of 1), and red as the global maximum enrichment value. Each cell corresponding to a genome coverage percentage value is colored based on its value with white as the minimum and green as the maximum. All annotations were accessed through the UCSC Genome Browser[1] except for nonsingleton mutations from Nextstrain[27] and homoplastic mutations from a prior study[18]. **c** Phylogenetic tree of the vertebrate CoV included in the alignment. Each leaf corresponds to a vertebrate CoV strain included in the vertebrate CoV. This tree was generated by pruning out SARS-CoV-2 genomes except the reference from the phylogenetic tree of the 119-way vertebrate CoV alignment obtained from the UCSC Genome Browser[1] ("Methods") and was plotted using Biopython[35]. SARS-CoV-2/Wuhan-Hu-1, the reference genome of the alignment, is at the top.

by PhastCons using the same alignments provided to ConsHMM in their ability to predict genes and regions of interest ("Methods"). When predicting bases overlapping genes or regions of interest within them, in most cases at least one of the conservation states achieves substantially greater precision at the same recall levels than PhastCons and PhyloP annotations (Supplementary Fig. 3). This suggests that when compared to existing constraint annotations based on the same alignments, ConsHMM conservation states capture additional biologically relevant information. Consistent with this, while some states have distinct distributions of PhastCons and PhyloP scores and fractions of constrained bases, many states have largely overlapping distributions of them (Supplementary Fig. 4).

**Conservation states' relationship to nonsingleton SARS-CoV-2 mutations observed in the pandemic.** We next investigated how the learned conservation states relate to nonsingleton SARS-CoV-2 mutations observed in the current pandemic (Fig. 4a, c). Specifically, we analyzed the state enrichment patterns for mutations observed at least twice in about 4000 SARS-CoV-2 sequences from GISAID (Global Initiative on Sharing All Influenza Data)[17]. To focus on reliable calls of mutations, we limited our analysis to nonsingleton mutations and masked genomic positions with known technical issues[14] ("Methods"). In the Sarbecovirus model, as expected, states with high probabilities that all strains align to and match SARS-CoV-2 (S17, S18) are significantly depleted of mutations observed in the current pandemic (0.6–0.7-fold enrichment; $P < 0.0001$) while several states (S6, S12, S19, S26, S28, S29) are significantly enriched for mutations (1.3–2.4-fold; $P < 0.001$).

The vertebrate CoV model's conservation states exhibit additional enrichment patterns for nonsingleton SARS-CoV-2 mutations. The model learns several states that are depleted of mutations with a minimum fold enrichment of 0.2 ($P < 0.0001$; V11), which is a stronger depletion than the minimum enrichment of 0.6 observed in the Sarbecovirus model. This is expected as the vertebrate CoV alignment contains a more diverse set of strains and is thus likely to capture deeper constraint than the Sarbecovirus alignment (Fig. 3c). Moreover, while the states significantly depleted of mutations in the Sarbecovirus model have high align and match probabilities for all strains (S17, S18), states significantly depleted of mutations in the vertebrate CoV model include not only an analogous state with high align and match probabilities for all vertebrate CoV (V27; 0.2-fold

enrichment; $P < 0.0001$), but also several states that have high align and match probabilities for only a specific subset of vertebrate CoV (0.2–0.4-fold; $P < 0.0001$; V10, V11). This subset excludes strains in a specific subtree in the phylogeny of CoV, largely consisting of CoV from avian hosts (Supplementary Fig. 5). This indicates that bases constrained among a specific subset of vertebrate CoV, which appear to have diverged in some of the avian CoV genomes, may be as important to SARS-CoV-2 as those constrained across all vertebrate CoV. In addition, the vertebrate CoV model learns states that are significantly enriched for mutations (1.5–1.8-fold; $P < 0.0001$; V3, V13, V20, V30). The enrichment patterns for nonsingleton mutations reported here are largely consistent when we include all observed mutations or control for the nucleotide composition of each base being mutated (Supplementary Fig. 6). These patterns are also largely consistent when we control for whether each mutation is intergenic, synonymous, missense, or nonsense, indicating that the observed state enrichment patterns are not simply driven by mutation type (Supplementary Fig. 6).

To understand the state annotation's relationship to positive selection, we next examined state enrichment patterns for homoplastic mutations (Fig. 4b, d). Specifically, we examined 198 stringently identified homoplastic mutations from a previous study[18]. These mutations were independently and repeatedly observed in separate SARS-CoV-2 lineages and are therefore more likely to be under positive selection than other mutations. State S6, which annotates bases with high align probability for all Sarbecoviruses, but high match probability specifically for bat CoV RaTG13 only, is enriched for homoplastic mutations (2.3-fold; $P < 0.001$). Similarly, state V13 is significantly enriched for homoplastic mutations (2.7-fold; $P < 0.001$), significantly more so than for nonsingleton mutations (1.5-fold; binomial $P < 0.05$). This state corresponds to bases that align to and match about a third of the vertebrate CoV, which excludes CoV with avian hosts and others. The state is also enriched for the nucleocapsid protein, particularly its dimerization and RNA-binding regions which are highlighted by UniProt[19] (14-, 16-, and 17-fold, respectively; $P < 0.0001$).

Notably, state S17, which has high align and match probabilities for all Sarbecoviruses, is strongly depleted of nonsingleton mutations and homoplastic mutations (0.7- and 0.6-fold enrichment, respectively; $P < 0.0001$). Interestingly, specific mutations that were previously suggested to be consequential to SARS-CoV-2 are also in this state. For example, in state S17 is a frequently

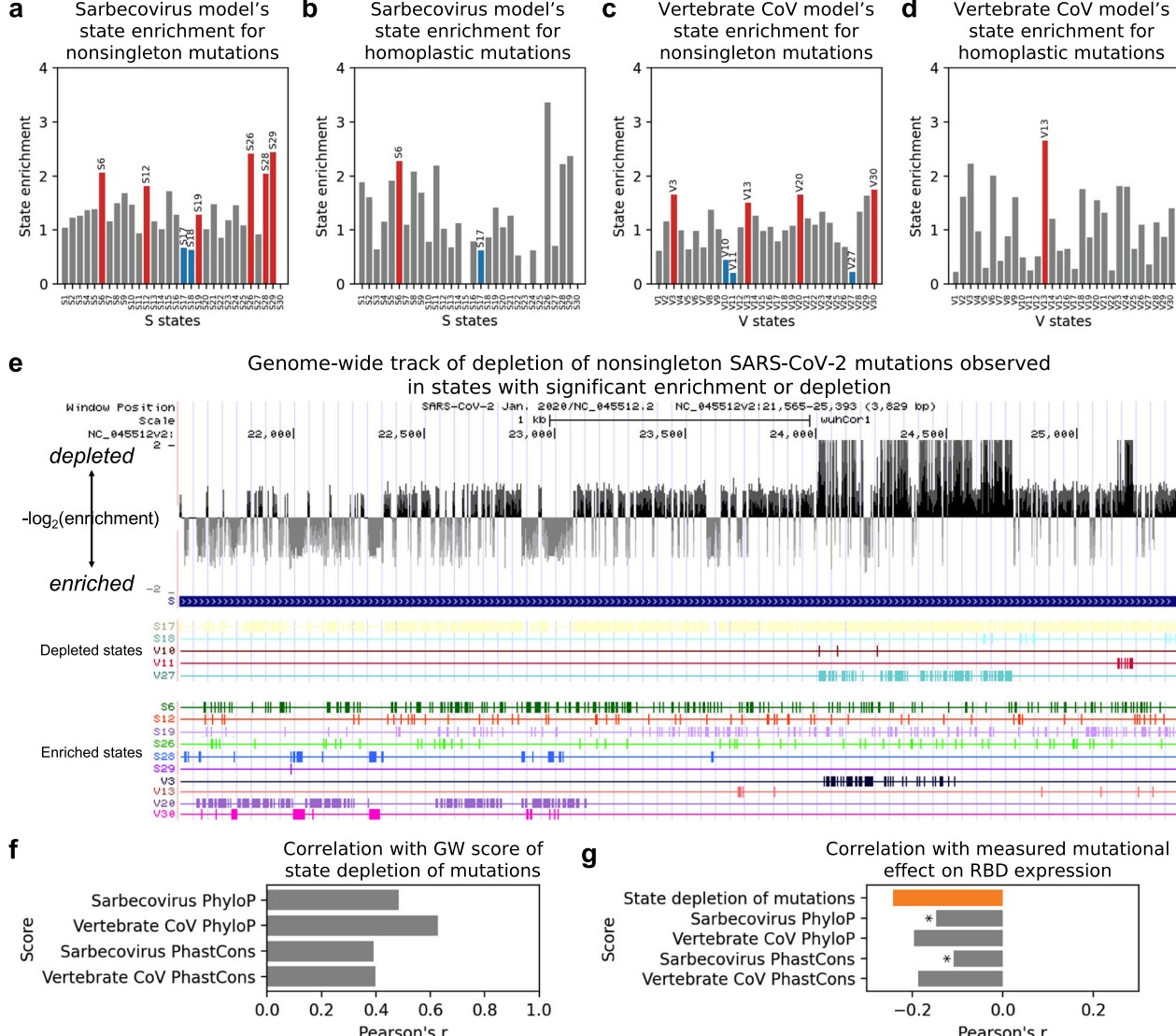

observed missense mutation (position 14408) in the coding region of RdRp that was previously suggested to contribute to worsening the virus's proofreading mechanism, making it easier for the virus to adapt and harder for its hosts to gain immunity[20]. The D614G mutation in the spike protein that was implicated to disrupt a Sarbecovirus-conserved residue[9] and result in increased infectivity[21] is also annotated by state S17. These occurrences of potentially consequential mutations in a state depleted of mutations are consistent with the notion that the state is experiencing negative selection and new mutations that do occur in the state are more likely to have stronger consequences than mutations introduced elsewhere. This depletion of potentially more consequential mutations is also seen with mutation type annotations, where 4% of all possible synonymous mutations are observed as nonsingleton mutations whereas only 0.3% of all possible nonsense mutations are observed as nonsingletons, reflecting their well-established difference in deleteriousness, though as noted above the conservation states show distinct enrichments for observed mutations even when conditioned on mutation type.

**Genome-wide tracks based on state depletion of SARS-CoV-2 mutations**. We next generated genome-wide tracks that reflect state depletion of mutations to highlight bases where new mutations are more likely to be consequential (Fig. 4e). Specifically, for each ConsHMM model, we generated a track that scores each genomic base by its state's statistically significant depletion or enrichment of nonsingleton mutations, reflecting the mutation frequency among bases that likely share a common evolutionary history. To merge distinct information captured by the two ConsHMM models, we also generated an integrated genome-wide track, where given two states from different ConsHMM models annotating a base of interest that are both either depleted or enriched for nonsingleton mutations we annotated the base with the state with stronger depletion or enrichment ("Methods").

We analyzed these tracks based on state depletion of mutations with respect to experimentally measured mutational effect on RBD from a previous study that conducted a deep mutational scanning of RBD[22]. The study specifically measured changes in RBD expression and binding affinity due to each possible amino acid change within RBD, where a positive value denoted increased expression or affinity and a negative value denoted decreased expression or affinity. We observe that all three tracks based on state depletion of mutations are negatively correlated with measured expression changes caused by single nucleotide mutations (Pearson's $r$: $-0.24\sim-0.18$, $P < 0.0001$; Fig. 4g and Supplementary Fig. 7c), which is consistent with our expectation

**Fig. 4 State enrichment patterns for nonsingleton mutations in the current pandemic and their relation to other annotations. a** Bar graph showing enrichment values of states S1–S30 learned from the Sarbecovirus sequence alignment for nonsingleton mutations ($n = 2201$; "Methods"). Red and blue bars correspond to states that enriched and depleted, respectively, with statistical significance after Bonferroni correction ("Methods"). Above each red or blue bar is the state ID. Grey bars correspond to states for which the enrichment was not statistically significant. Nonsingleton mutations were identified from Nextstrain mutations[27]. **b** Similar to **a** but showing state enrichment values for homoplastic mutations ($n = 198$) instead of nonsingleton mutations in states S1–S30. Homoplastic mutations are mutations independently and repeatedly observed in separate SARS-CoV-2 lineages and were previously stringently identified through maximum parsimony tree reconstruction and homoplasy screen using thousands of SARS-CoV-2 sequences[18]. **c** Similar to **a** but showing state enrichment values of states V1–V30 learned from the vertebrate CoV sequence alignment instead of states S1–S30. **d** Similar to **b** but showing state enrichment values of states V1–V30 learned from the vertebrate CoV sequence alignment instead of states S1–S30. **e** Genome browser view of gene *S* with an integrated score of depletion of nonsingleton mutations in conservation states derived from both ConsHMM models and annotations of states from which the score is generated. Top row with black and grey vertical bars corresponds to the score, which is a negative $\log_2$ of the fold enrichment value of a state selected from one of the ConsHMM models that annotates a given base and is statistically significantly enriched or depleted of nonsingleton mutations at a genome-wide level ("Methods"). The following rows correspond to the states with significant enrichment or depletion. **f** Bar graph showing correlation between our genome-wide (GW) score of state depletion of mutations shown in **e** and four sequence constraint scores listed along the y-axis. The sequence constraint scores were based on either the Sarbecovirus or vertebrate CoV sequence alignment provided to ConsHMM using either PhastCons or PhyloP as the scoring method ("Methods"). Similar plots using scores of mutation depletion in states from each ConsHMM model separately instead of both models together are shown in Supplementary Fig. 7a, b. **g** Bar graph showing correlation between measured mutational effect on RBD expression and five scores which include our genome-wide score based on state depletion of mutations and the four sequence constraint scores from **f**. Correlation computed with our state-based score is shown in orange. Correlations computed with sequence constraint scores are shown in grey. All correlations were statistically significant after Bonferroni correction ("Methods"). Asterisk is shown next to a grey bar if its corresponding correlation was statistically significantly different than the correlation with our state-based score based on Zou's confidence interval test[32] with Bonferroni correction ("Methods"). The null hypothesis is rejected if the confidence interval (99.6% after correction) of a difference between two correlations excludes 0. The confidence intervals corresponding to the top and bottom asterisks are ($-0.18$, $-0.01$) and ($-0.22$, $-0.05$), respectively. Mutational effect on RBD expression was measured by a study that conducted a deep mutational scanning of 3,819 nonsynonymous mutations in RBD[22]. To compute the correlations, we restricted to the 1,215 mutations that were caused by single nucleotides and free of experimental measurements that were not determined (n.d.). A positive value indicates increased expression due to mutation and a negative value indicates decreased expression. An extended version of this plot that includes two genome-wide scores based on mutation depletion in states from each ConsHMM model separately is shown in Supplementary Fig. 7c.

that mutations at bases depleted of observed mutations in general are likely to be more deleterious than other mutations. Furthermore, we observe significant negative correlation between the track based on the vertebrate CoV state annotations and binding affinity changes (Pearson's $r$: $-0.12$; $P < 0.0001$; Supplementary Fig. 7d).

We further compared the state-based tracks to four sequence constraint scores that were learned from either alignment provided to ConsHMM using PhastCons[4] or PhyloP[5] ("Methods"). Specifically, we examined the constraint scores' correlation with our tracks based on state depletion of mutations and also with measured mutational effect on RBD expression and binding affinity. The constraint scores are moderately correlated with our state-based genome-wide tracks (Fig. 4f and Supplementary Fig. 7a, b; Pearson's $r$: 0.25–0.63). For the evaluation on measured mutational effect on RBD expression, we see a statistically significant difference with constraint scores, with two out of four constraint scores having statistically significantly weaker correlation than our tracks' correlations with the mutational effect (Fig. 4g and Supplementary Fig. 7c; $P < 0.004$; "Methods").

Overall, our genome-wide tracks based on significant depletion of mutations in conservation states show expected agreement with measured mutational effect. This suggests that our genome-wide tracks based on depletion of mutations could help prioritize mutations with strong impact on the virus's protein expression and binding affinity or potentially other functionalities, but we note that this analysis does not provide direct evidence for other parts of the genome or other phenotypes of the virus.

## Discussion
Here we applied a comparative genomics method ConsHMM to two sequence alignments of CoV, one consisting of Sarbecoviruses that infect human and bats and the other consisting of a more

diverse collection of CoV that infect various vertebrates. The conservation states learned by ConsHMM capture combinatorial and spatial patterns in the multi-strain sequence alignments. The states show associations with various other annotations not used in the model learning. The conservation state annotations are complementary to constraint scores, as they capture a more diverse set of evolutionary patterns of bases aligning and matching, enabling one to group genomic bases by states and study each state's functional relevance. Identifying patterns of conservation across different strains can be important potentially for understanding the relative pathogenicity of different coronaviruses and cross-immunity from prior infections[23–25]. It should be noted, however, that ConsHMM does not consider where bases in the reference strain align to in non-reference strains and is therefore not expected to capture large-scale rearrangements.

We showed that certain conservation states are strongly enriched or depleted of nonsingleton SARS-CoV-2 mutations. Based on this information, we generated three genome-wide tracks that can be used to prioritize mutations of potentially greater consequence based on evolutionary information of the Sarbecovirus and vertebrate CoV alignments. We note that these tracks are generated in a transparent way directly from the fold enrichment values for nonsingleton mutations observed in the conservation states. Overall, we expect the two sets of conservation state annotations along with these tracks based on state depletion of mutations to be resources for locating bases with distinct evolutionary patterns and analyzing mutations that are currently accumulating among SARS-CoV-2 sequences.

## Methods
**Sequence alignments**. We obtained the 44-way Sarbecovirus sequence alignment from the UCSC Genome Browser[1] (http://hgdownload.soe.ucsc.edu/goldenPath/wuhCor1/multiz44way/). We obtained the vertebrate CoV sequence alignment by first downloading the 119-way vertebrate CoV sequence alignment from the UCSC

Genome Browser (http://hgdownload.soe.ucsc.edu/goldenPath/wuhCor1/multiz119way/) and then removing SARS-CoV-2 sequences from the alignment, except the reference sequence, wuhCor1. This resulted in 56 CoV aligned against the reference. Both sequence alignments were generated by the alignment tool Multiz[26].

**External annotations**. Mutations found in SARS-CoV-2 sequences were point mutations identified by Nextstrain[27] (accessed on Sept 7, 2020) from sequences available on GISAID[17]. For our analysis, to minimize putative false calls we filtered out mutations if their ancestral alleles did not match the reference genome used by Nextstrain, MN908947.3, such as C > T at a base where T is the reference allele. All the other annotations, including the annotations of genes, codons, and UniProt protein products and regions of interest, were accessed through the UCSC Genome Browser (accessed on Sept 7, 2020)[1].

**Learning ConsHMM conservation states and choice of number of states**. Given the two input sequence alignments, we first learned multiple ConsHMM models from each alignment with varying numbers of states ranging from 5 to 100 with increments of 5 and then chose a number of states that is applicable to both alignments. Specifically, we aimed to find a number of states that results in states few enough to be relatively easy to interpret, but specific enough to capture distinct patterns in the alignment data.

To do so, for each model, we considered whether the model's states had sufficient coverage of the genome to avoid having states that annotate too few bases. We additionally considered whether the model's states exhibited distinct emission parameters to ensure that they were different enough to capture distinct patterns in the alignment data. Lastly, we considered whether the model's states showed distinct enrichment patterns for external annotations of genes, protein domains, and mutations in SARS-CoV-2 to ensure that the different states annotate bases with potentially different biological roles. As a result, we chose 30 as the number of conservation states for both the Sarbecovirus and vertebrate CoV ConsHMM models because the resulting states were sufficiently distinct in their emission parameters and association with external annotations and most of the states covered more than 0.5% of the genome.

**PhastCons and PhyloP scores**. We obtained the 44-way PhastCons and PhyloP scores learned from the Sarbecovirus sequence alignment from the UCSC Genome Browser (http://hgdownload.soe.ucsc.edu/goldenPath/wuhCor1/). We additionally used the PHAST software[28] to learn PhastCons and PhyloP scores from the vertebrate CoV sequence alignment that we generated from the 119-way alignment as described above. To do so, we first ran 'tree_doctor' to prune out SARS-CoV-2 sequences except the reference from the phylogenetic tree generated for the 119-way alignment. We then followed the procedure used to generate the 44-way and 119-way scores as described on the UCSC Genome Browser. Specifically, to learn the vertebrate CoV PhastCons score, we used the following arguments to run 'phastCons': --expected-length 45 --target-coverage 0.3 --rho 0.3. To learn the vertebrate CoV PhyloP score, we used the following arguments to run 'phyloP': --wig-scores --method LRT --mode CONACC.

**Masking bases**. For all but one downstream analysis, we masked problematic genomic positions listed in the UCSC Genome Browser track 'Problematic Sites' (accessed on Sept 7, 2020) as they are likely affected by sequencing errors, low coverage, contamination, homoplasy, or hypermutability[14,29,30]. As a result, we masked 228 bases, analyzing 29,675 out of 29,903 bases (99.2%). The one exception was when we computed state enrichment for homoplastic mutations from a prior study[18]. For this analysis only, we masked all problematic positions except for those described as homoplastic or highly homoplastic. As a result, we masked 175 bases instead of 228 bases, analyzing 29,728 bases (99.4%).

**Fold enrichments for external annotations**. When computing fold enrichments for annotations of genes, positions within codons, and regions of interest, we considered whether a genomic base is annotated or not by the external annotations. To compute the fold enrichment for each external annotation and each state, we divided the fraction of the state's bases in the external annotation out of all bases in the state by the fraction of bases in the external annotation genome-wide. Because multiple mutations could be observed in the same genomic base, when computing fold enrichments for mutations, we first generated all possible point mutations in the SARS-CoV-2 genome and then considered whether each of the possible mutations was observed or not. Thus, to compute fold enrichment for mutations in an external annotation for each state, we divided the fraction of observed mutations in the external annotation among possible mutations occurring at bases in the state by the fraction of observed mutations in the external annotation out of all possible mutations genome-wide. We defined nonsingleton mutations as mutations observed in at least two SARS-CoV-2 sequences. For homoplastic SARS-CoV-2 mutations, we used all 198 mutations reported in a prior study[18]. For all fold enrichment values, we also conducted a two-sided binomial test to report statistical

significance. We applied a Bonferroni correction by setting the significance threshold to 0.05 divided by 30, the number of states.

**Correction of state enrichments for SARS-CoV-2 mutations by nucleotide composition or mutation type**. To show that the conservation state fold enrichment values for nonsingleton mutations are not simply driven by nucleotide composition or mutation type (i.e. intergenic, synonymous, missense, nonsense), we corrected state enrichment values by nucleotide composition or mutation type as follows. To control for nucleotide composition, for each nucleotide $i$, we first computed the genome-wide fraction $f_i$ of observed nonsingleton mutations out of all possible mutations with nucleotide $i$ as the reference base. Then for each state and for each nucleotide $i$, we multiplied the genome-wide fraction $f_i$ and the number of possible mutations in the state with nucleotide $i$ as the reference base. For each state, we summed up these values across the nucleotides to obtain the expected number of nonsingleton mutations based on nucleotide composition. Finally, the enrichment corrected by nucleotide composition for each state was computed as the ratio of actual and expected number of observed nonsingleton mutations.

Similarly, to control for mutation type, for each type $j$, we computed the genome-wide fraction $f_j$ of observed nonsingleton mutations out of all possible mutations belonging to mutation type $j$. Then for each state and for each mutation type $j$, we multiplied the genome-wide fraction $f_j$ with the number of possible mutations in the state belonging to mutation type $j$. We then followed the same procedure as above.

**Identifying bases unique to pathogenic human CoV and missing in less pathogenic human CoV**. We first identified bases annotated by state V14, which corresponds to high align probability for pathogenic human CoV (SARS-CoV, MERS-CoV) and low align probability for less pathogenic human CoV (OC43, HKU1, 229E, and NL63) in the vertebrate CoV sequence alignment. Among these bases, we then identified bases that appeared among all pathogenic human CoV but missing in all less pathogenic human CoV in an alignment of 944 human CoV sequences generated by a prior study. All the 944 sequences come from the seven human CoV including SARS-CoV-2[16].

**Precision-recall analysis for recovery of annotated genes and regions of interest**. For each NCBI gene[31] or UniProt region of interest[19], we predicted bases in each state from both models to be in the gene or region and computed precision and recall, resulting in 60 pairs of precision and recall values. Similarly, we predicted all bases annotated as a PhastCons element[4] to be in each gene or region and computed precision and recall. With PhastCons and PhyloP scores[5], we computed precision-recall curve for predicting the bases in each gene or region using each score.

**Generating browser tracks of depletion of nonsingleton SARS-CoV-2 mutations**. Based on the procedure of computing state enrichment of SARS-CoV-2 mutations, for each ConsHMM model, we selected states that exhibited statistically significant enrichment or depletion of nonsingleton mutations at a binomial test $p$-value threshold of 0.05 after Bonferroni correction. To generate a track for each ConsHMM model, we scored each base overlapping any of the selected states in the model with $-\log_2(v)$ where $v$ is the fold enrichment value of the state annotating the base, such that stronger depletion of mutations corresponded to a higher score above 0 and stronger enrichment to a lower score below 0. Bases not annotated by any of the selected states were assigned a score of 0.

We generated an integrated track of mutation depletion in states from both ConsHMM models as follows. If a base was annotated with two states with statistically significant enrichment or depletion of nonsingleton mutations, each from different ConsHMM models, and the two states agreed in the enrichment direction (enriched or depleted), we annotated the base with the $-\log_2(v)$ from the state that had a higher absolute value of $-\log_2(v)$. If a base was annotated with two of the selected states, but the states disagreed in the enrichment direction, we annotated the base with a score of 0. If a base was annotated by one state with statistically significant enrichment or depletion of nonsingleton mutations, we annotated the base with the $-\log_2(v)$ value from that state. Bases not annotated by any of the selected states were assigned a score of 0.

**Comparing correlation to mutational effect on RBD expression and binding affinity**. For each of the three aforementioned genome-wide tracks based on state depletion of mutations, we computed its Pearson's $r$ with mutational effect on RBD expression measured by a previous study[22]. For each of the four sequence constraint scores, we also computed its correlation with mutational effect on RBD expression and then compared it to the correlations computed using our genome-wide tracks, using Zou's confidence interval test[32] implemented in the R package cocor[33]. The four sequence constraint scores included PhyloP and PhastCons scores learned from either the Sarbecovirus or vertebrate CoV alignment. When reporting the significance of correlations, we applied a Bonferroni correction by setting the significance threshold to 0.05 divided by 7, the total number of

computed correlations. When comparing correlations using Zou's confidence interval test, we compared a state-based track's correlation to a constraint score's correlation if at least one of the two correlations was negative and statistically significant and applied a Bonferroni correction by setting the confidence level to $1 - 0.05/n$ where $n$ is the total number of pairwise comparisons, which was at most 12. The same procedure was applied to compute correlations with measured mutational effect on RBD binding affinity.

**Statistics and reproducibility**. All statistical tests performed are described in detail above. In general, Bonferroni correction was applied and a threshold of 0.05 was used to discern statistical significance.

**Reporting summary**. Further information on research design is available in the Nature Research Reporting Summary linked to this article.

## Data availability
ConsHMM conservation state annotation based on the Sarbecovirus and vertebrate CoV alignments are available at https://github.com/ernstlab/ConsHMM_CoV/. Track annotations of depletion of mutations observed in conservation states from both Sarbecovirus and vertebrate CoV ConsHMM models or each model are available from the same URL. All annotations are also included in Supplementary Data 1. Source data for Figs. 2a–b, 3a–b, 4a–d, f–g, and Supplementary Fig. 7 are provided in Supplementary Data 2.

## Code availability
We used ConsHMM v1.1 obtained from https://github.com/ernstlab/ConsHMM/.

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

## Acknowledgements
We gratefully acknowledge all those who contributed to generating and sharing their SARS-CoV-2 sequence data via GISAID. We thank those at Nextstrain.org who made their processed mutation data publicly available. We also thank Adriana Arneson for assistance on using ConsHMM. We thank Sriram Sankararaman for comments on the manuscript. This research was supported by the UCLA David Geffen School of Medicine – Eli and Edythe Broad Center of Regenerative Medicine and Stem Cell Research Award Program, the US National Institutes of Health (DP1DA044371), and the National Science Foundation (2125664).

## Author contributions
S.K. and J.E. analyzed the results and wrote the paper.

## Competing interests
The authors declare no competing interests.
