## [Peer Review File · Communications Biology]

Reviewers' comments:

Reviewer #1 (Remarks to the Author):

The manuscript "Single-nucleotide conservation state annotation of the SARS-CoV-2 genome" by Soo Bin Kwon and Jason Ernst offers a novel and highly informative approach to the problem of assessing the stability of the SARS-CoV-2 genome. In the manuscript, the authors use genome-wide alignments of (i) Sarbecoviruses and (ii) coronaviruses, to explore conserved and mutation-prone regions of the coronavirus genome. They do this using a HMM that decomposes the alignment signal to 30 different states that captures different biological and functional properties of the genome architecture. This research will be of interest to a large number of people working with the monitoring of SC2 evolution, functional or experimental genomics and vaccine design. I found the manuscript to be very well written and clever in its non-traditional approach to this tricky subject.

I have some specific comments, questions and suggestions, listed below:

1. The alignments used does not appear to have been produced by the authors themselves, and the way they were created is a bit unclear. I have tried looking at the 44-way alignment of the Sarbecoviruses, which appears to have been generated by the program multiz. In the description of this program it is clearly stated that stuff like inversions and duplications will be overlooked, and the program assumes that all blocks will occur in the same order and orientation in all genomes. With recombination being a significant factor here, do the authors consider it likely that these types of mutations could impact what we see?
2. In the introduction: "ConsHMM does not explicitly use any phylogenetic information. This is fitting for annotating virus genomes such as SARS-CoV-2 since frequent recombination among viruses makes it difficult to build an accurate tree." - Although true, this is slightly misleading. As far as I'm aware there is not much conclusive evidence about recombination within SC2 itself, and most apparent recombinations have been shown to be contaminations or lab artefacts. Due to the overwhelming amount of published articles on SC2 I might have missed something here, but I would like the authors to cite evidence of recombination within SC2 specifically if they maintain that it not only exists but is frequent. With that said, recombination is very likely to have played a role in its evolution and in coronaviruses in general, so the statement is true when applied to the subgenus and family levels.
3. The number of emission states chosen was 30. I am intrigued by the essentially "blind" approach to decomposing the alignment signal. Although the authors' strategy is well explained here, the number of states chosen remains a qualitative judgement that depends on how they appear to capture biological properties. In the future it would be very interesting to see if this process could become more automated, and the learned attributes of each state (similar to supplementary tables 1 and 2) could be inferred by the program itself. Note that I am NOT suggesting that the authors do this for this manuscript. I am just highlighting that as something that I would be very excited to see in the future.
4. "For our analysis, we filtered out mutations if their ancestral alleles did not match the reference genome used by Nextstrain" - I don't understand what is meant here. Is this to exclude multiple, serial mutations? (E.g. A -> T -> C. Here C is the observed mutation and A is the reference, but the ancestral of C is T so it gets excluded?)
5. Figures 2 and 3, panel (b). In both figures, "orf1ab" is listed three times as a column. Also, the accession numbers do not always correspond to the annotation. For example, the second column is "orf1ab (YP_009725295.1)" However, in Genbank, this accession is orf1a. The third column however HAS an accession that points to orf1ab, but the numbers here (coverage etc) seem to correspond to a smaller part of the polyprotein.
6. Figure 3 - panel (c). The virus is listed as "Bat Cov TG13" rather than RaTG13. (Very minor, but worth fixing)
7. Figure 4 - There is something strange here. Is the annotation in panel f reversed? For example, from (c) we know that V27 is strongly non-enriched for non-singleton mutations. In other words, it captures a very stable part of the genome. However, in (f) it is marked as an "enriched state" on the left, but the top bar plot indicates depletion in the region associated with V27.

Reviewer #2 (Remarks to the Author):

This study quantified conservation within the SARS-CoV-2 genome at nucleotide resolution by comparison to other coronavirus (CoV) species. The aim is to provide a resource that others can use to interpret the possible functional consequences of mutations within the SARS-CoV-2 genome. With novel strains emerging all over the globe, this is clearly a matter of enormous practical significance.

The authors use a method that they developed called ConsHMM, which uses a multivariate hidden Markov model to classify each nucleotide in the SARS-CoV-2 genome into 1 of many discrete states. They apply this to an alignment of 44 Sarbecoviruses and to an alignment of 119 CoV representing a broader phylogenetic range. The model learned numerous distinct conservation states, some corresponding to results of previous analyses, some intriguing in that they are novel patterns that may be the product of natural selection, and some that are more difficult to interpret. Finally, the authors show that some conservation states are enriched or depleted among SARS-CoV-2 strains worldwide. This information was used to generate a genome track whose goal is to rank mutations for likely functional significance.

Overall, this study presents a creative and novel approach to understanding CoV genome evolution. I am intrigued and impressed by the findings, but not entirely convinced that the methods can actually identify biologically meaningful patterns. That said, the only way to find out is to make these results known to the research community so that they can be tested empirically.

Main concerns:

1. The authors argue in the Introduction that an advantage of using an HMM-based approach like ConsHMM is that it "does not explicitly use any phylogenetic information." They base this on the claim that recombination "makes it difficult to build an accurate tree" when studying the evolution of coronavirus genomes. This is a bit disingenuous. ConsHMM uses whole genome alignments that were generated using an algorithm that builds a phylogenetic tree prior to optimizing the alignment. If building a phylogenetic tree from CoV genomes is as difficult as the authors note, then the input files they analyzed are an inherent weakness that needs to be noted ("garbage in, garbage out"). Related to this point, while it is problematic to build an accurate tree for the entire CoV genome, it is not problematic to do so for segments of the genome. Indeed, identifying discordant topologies in different regions of the genome can provide enormously useful information, such as likely sites of recombination within CoV genomes. Several published studies have taken advantage of this approach and gleaned many insights from it. Phylogenetically informed approaches are not perfect, of course; the question is what insights HMM-based approaches provide that they cannot. Given that most readers will be more familiar with phylogenetically-based approaches, it would be very helpful to lay this out clearly in the manuscript.

2. While ConsHMM can indeed "capture distinct patterns in the input alignment data" (Introduction), the biological interpretation of these patterns is less clear. For instance, the first section of the Results (based on 44 Sarbecoviruses) presents interpretations for just 4 out of 30 states. Do the other 26 states have plausible biological interpretations? And are the four states that are discussed ones that would not be detected using phylogeny-based approaches? (By "interpretation" I mean insight, not descriptions such as those in Supplementary Table 1.)

3. With phylogeny-based methods, the biological interpretation of a result is often quite clear because there is a straightforward model for differences as the product of mutation, selection, drift, and recombination. In other words, the metrics map onto well understood processes. With the HMM approach, it feels more like interpreting a Rorschach blot. Given that the goal of the present study is to create a resource, the more the authors can do to guide interpretation the more effective their results will be. Experiments take a lot of money, effort and time, making a "blind" priority score from the browser track will be much less compelling than a mental model based on a biological mechanism.

None of this is to say HMM-based approaches should not be used or even that they are inferior to conventional methods that are phylogenetically-based. To the contrary, I would very much like to

see this study published and, more generally, the potential of HMM-based approaches to be more thoroughly investigated.

Minor issues:

1. It seems odd to present new results and comment on them in the Introduction (third paragraph). Indeed, Figure 1 is referenced only in the Introduction and not in the Results section, even though it presents new results.

2. When discussing HMM states, the narrative is often nonspecific (e.g., page 2 "One state", "another state", etc.). When discussing specific states please refer to them by number so that readers can find them in the tables and figures.

3. It would be helpful to have a little more information about masking bases (page 11). Some of criteria that UCSC uses to flag "Problematic Sites" are biological processes with a direct impact on understanding function (homoplasy, hypermutability), so masking bases can lead to the loss of important information. How many bases were masked in the various analyses? The effort by the authors to re-insert bases that were flagged specifically due to homoplasy for one analysis is much appreciated. What fraction of the overall flagged bases was this and much of an impact did this have on the results? Providing this information will give at least some limited insight into the effects of masking and how it might bias results.

4. The comparison of PhyloP and ConsHMM predictions of mutational effect (page 8) is much appreciated. While the results are suggestive, they are extremely limited in scope. In other words, only one aspect of function (expression) was tested for only one region of one protein (RBD). Please provide appropriate caveats when discussing these results.

5. Another point about mutational effect is that readers need to understand exactly what the effect is. The manuscript says "expression" but study that is referenced primarily investigated the effects on Spike-ACE2 binding. Please clarify so that readers understand exactly what mutational effects were used to compare PhyloP and ConsHMM predictions.

Reviewer #1 (Remarks to the Author):

The manuscript "Single-nucleotide conservation state annotation of the SARS-CoV-2 genome" by Soo Bin Kwon and Jason Ernst offers a novel and highly informative approach to the problem of assessing the stability of the SARS-CoV-2 genome. In the manuscript, the authors use genome-wide alignments of (i) Sarbecoviruses and (ii) coronaviruses, to explore conserved and mutation-prone regions of the coronavirus genome. They do this using a HMM that decomposes the alignment signal to 30 different states that captures different biological and functional properties of the genome architecture. This research will be of interest to a large number of people working with the monitoring of SC2 evolution, functional or experimental genomics and vaccine design. I found the manuscript to be very well written and clever in its non-traditional approach to this tricky subject.

We thank the reviewer for the summary and the positive comments.

I have some specific comments, questions and suggestions, listed below:

We thank the reviewer for the constructive questions and feedback that have improved this work. We describe below how we addressed them.

1. The alignments used does not appear to have been produced by the authors themselves, and the way they were created is a bit unclear. I have tried looking at the 44-way alignment of the Sarbecoviruses, which appears to have been generated by the program multiz. In the description of this program it is clearly stated that stuff like inversions and duplications will be overlooked, and the program assumes that all blocks will occur in the same order and orientation in all genomes. With recombination being a significant factor here, do the authors consider it likely that these types of mutations could impact what we see?

We apologize that it was not made clear how the input alignments were created. We now specify this information at the end of section 'Sequence alignments' in Methods as follows:

"Both sequence alignments were generated by alignment tool Multiz."

We want to clarify that ConsHMM considers which SARS-CoV-2 bases align and match to which sequences in the input alignment. It therefore does not explicitly differentiate recombination and mutation. Although Multiz may not be able to capture all recombination events, it could still capture some in its alignments. Therefore some of the patterns in the Multiz alignment captured by ConsHMM could be attributed to recombination and some to mutation. As mentioned in the main text, for example, state V20 has high align and match probabilities for CoV from bat and pangolin and is enriched for a region where a recombination event between a bat CoV and a pangolin CoV might have occurred¹. We note that because Multiz assumes that all alignment blocks occur in the same order in all genomes, large-scale rearrangements would not be captured by ConsHMM as it considers whether bases in the reference genome align

to other genomes but not where the bases align to in other genomes. To raise this point, we added the following sentence to Discussion:

“It should be noted, however, that ConsHMM does not consider where bases in the reference strain align to in non-reference strains and is therefore not expected to capture large-scale rearrangements.”

2. In the introduction: "ConsHMM does not explicitly use any phylogenetic information. This is fitting for annotating virus genomes such as SARS-CoV-2 since frequent recombination among viruses makes it difficult to build an accurate tree." - Although true, this is slightly misleading. As far as I'm aware there is not much conclusive evidence about recombination within SC2 itself, and most apparent recombinations have been shown to be contaminations or lab artefacts. Due to the overwhelming amount of published articles on SC2 I might have missed something here, but I would like the authors to cite evidence of recombination within SC2 specifically if they maintain that it not only exists but is frequent. With that said, recombination is very likely to have played a role in its evolution and in coronaviruses in general, so the statement is true when applied to the subgenus and family levels.

We apologize for the confusion and thank the reviewer for pointing this out. By “recombination among viruses”, we meant recombination among coronaviruses in general and not among SARS-CoV-2 strains. We also realized that because ConsHMM is useful even when an accurate phylogenetic tree can be built that this sentence was unnecessary and confusing. We therefore removed the sentence on recombination and accuracy of phylogenetic trees.

3. The number of emission states chosen was 30. I am intrigued by the essentially "blind" approach to decomposing the alignment signal. Although the authors' strategy is well explained here, the number of states chosen remains a qualitative judgement that depends on how they appear to capture biological properties. In the future it would be very interesting to see if this process could become more automated, and the learned attributes of each state (similar to supplementary tables 1 and 2) could be inferred by the program itself. Note that I am NOT suggesting that the authors do this for this manuscript. I am just highlighting that as something that I would be very excited to see in the future.

We thank the reviewer for the feedback. We agree with the reviewer that this would be an interesting direction to pursue in future work.

4. "For our analysis, we filtered out mutations if their ancestral alleles did not match the reference genome used by Nextstrain" - I don't understand what is meant here. Is this to exclude multiple, serial mutations? (E.g. A -> T -> C. Here C is the observed mutation and A is the reference, but the ancestral of C is T so it gets excluded?)

We apologize for the confusion and thank the reviewer for the question. Our filtering strategy was primarily to exclude mutations called by Nextstrain as there is reason to

suspect they may be artifacts from building a phylogenetic tree of SARS-CoV-2 sequences and not true mutations. We filtered out 83 mutations, 78 of which came from serial changes such as 'A -> T -> A' where two serial mutations result in returning to the reference allele. We exclude the second mutation as it is likely an artifact of tree reconstruction. Among the 83 excluded mutations, five came from multiple serial changes involving two derived alleles such as 'A -> T -> C'. We exclude the second mutation here as well. Including the second mutation is unlikely to alter our analyses as such cases are rare and the first mutation is still considered in the analysis. To clarify this, we edited the main text to read as follows:

“For our analysis, we filtered out mutations if their ancestral alleles did not match the reference genome used by Nextstrain, MN908947.3, to minimize putative false positive calls such as C>T at a base where T is the ancestral allele.”

5. Figures 2 and 3, panel (b). In both figures, "orf1ab" is listed three times as a column. Also, the accession numbers do not always correspond to the annotation. For example, the second column is "orf1ab (YP_009725295.1)" However, in Genbank, this accession is orf1a. The third column however HAS an accession that points to orf1ab, but the numbers here (coverage etc) seem to correspond to a smaller part of the polyprotein.

We thank the reviewer for pointing this out. We have revised both figures so that the accession numbers correctly match the annotation. Instead of three “orf1ab”s, we now have “orf1a” and “orf1ab”, which is consistent with NCBI’s Gene track on UCSC Genome Browser.

6. Figure 3 - panel (c). The virus is listed as "Bat Cov TG13" rather than RaTG13. (Very minor, but worth fixing)

We thank the reviewer for pointing this out. This virus is now listed as “Bat CoV RaTG13” in **a** and **c** of **Figure 3**.

7. Figure 4 - There is something strange here. Is the annotation in panel f reversed? For example, from (c) we know that V27 is strongly non-enriched for non-singleton mutations. In other words, it captures a very stable part of the genome. However, in (f) it is marked as an "enriched state" on the left, but the top bar plot indicates depletion in the region associated with V27.

We thank the reviewer for pointing this out. The labels were switched by accident. This has been fixed now in the revised manuscript.

Reviewer #2 (Remarks to the Author):

This study quantified conservation within the SARS-CoV-2 genome at nucleotide resolution by comparison to other coronavirus (CoV) species. The aim is to provide a resource that others can use to interpret the possible functional consequences of mutations within the SARS-CoV-2 genome. With novel strains emerging all over the globe, this is clearly a matter of enormous practical significance.

The authors use a method that they developed called ConsHMM, which uses a multivariate hidden Markov model to classify each nucleotide in the SARS-CoV-2 genome into 1 of many discrete states. They apply this to an alignment of 44 Sarbecoviruses and to an alignment of 119 CoV representing a broader phylogenetic range. The model learned numerous distinct conservation states, some corresponding to results of previous analyses, some intriguing in that they are novel patterns that may be the product of natural selection, and some that are more difficult to interpret. Finally, the authors show that some conservation states are enriched or depleted among SARS-CoV-2 strains worldwide. This information was used to generate a genome track whose goal is to rank mutations for likely functional significance.

Overall, this study presents a creative and novel approach to understanding CoV genome evolution. I am intrigued and impressed by the findings, but not entirely convinced that the methods can actually identify biologically meaningful patterns. That said, the only way to find out is to make these results known to the research community so that they can be tested empirically.

We thank the reviewer for the summary and the positive comments. We also thank the reviewer for the constructive comments and address them as described below.

Main concerns:

1. The authors argue in the Introduction that an advantage of using an HMM-based approach like ConsHMM is that it "does not explicitly use any phylogenetic information." They base this on the claim that recombination "makes it difficult to build an accurate tree" when studying the evolution of coronavirus genomes. This is a bit disingenuous. ConsHMM uses whole genome alignments that were generated using an algorithm that builds a phylogenetic tree prior to optimizing the alignment. If building a phylogenetic tree from CoV genomes is as difficult as the authors note, then the input files they analyzed are an inherent weakness that needs to be noted ("garbage in, garbage out"). Related to this point, while it is problematic to build an accurate tree for the entire CoV genome, it is not problematic to do so for segments of the genome. Indeed, identifying discordant topologies in different regions of the genome can provide enormously useful information, such as likely sites of recombination within CoV genomes. Several published studies have taken advantage of this approach and gleaned many insights from it. Phylogenetically informed approaches are not perfect, of course; the question is what insights HMM-based approaches provide that they cannot. Given that most readers will more familiar with phylogenetically-based approaches, it would be very helpful to lay this out clearly in the manuscript.

We thank the reviewer for raising these points. We agree that ConsHMM is not entirely independent from the phylogenetic information present in the input alignments. However, ConsHMM in its modeling does not make any explicit assumption about phylogenetic relationships among the sequences. To clarify this point, we revised the end of the 2nd paragraph of the Introduction to now read:

“Apart from the input alignments which were generated using phylogenetic trees, ConsHMM does not explicitly use any phylogenetic information and therefore does not take any strict assumptions on the phylogenetic relationship among sequences. This allows ConsHMM to be more flexible in capturing various patterns within alignments than the more commonly used comparative genomics approaches that define a single constraint score based on phylogenetic modeling. Previous work applying ConsHMM to multi-species alignment of other genomes have shown that the conservation states learned by ConsHMM capture various patterns in the alignment overlooked by previous methods and are useful for interpreting DNA elements and phenotype-associated variants^{6,8}.”

To better highlight the strength of ConsHMM, we also now elaborate on the limitation of phylogenetically-based approaches in the previous paragraph as follows:

“While existing systematic annotations that quantify sequence constraint from alignments^{4,5} are informative, they reduce the information in the underlying alignment to a single univariate or binary value and thus are limited in the information they convey. Additional information about patterns of which sequences align to and match the SARS-CoV-2 genome at each bases may be useful in analyzing the SARS-CoV-2 genome and mutations⁶.”

Based on the reviewer’s comment, we also realized that the point about the difficulty of building an accurate phylogenetic tree as a motivation to apply ConsHMM was confusing and unnecessary. Even when the phylogenetic tree is accurate, we believe that ConsHMM is still useful. We therefore removed the sentence that discusses how ConsHMM is fitting for viral genomes because recombination among viruses makes it difficult to build an accurate tree.

Our initial paper on the ConsHMM method² had an extensive analysis of this modeling approach and comparisons to genome annotations from phylogenetic based constraint scores and elements, demonstrating that ConsHMM conservation states provide complementary information to existing sequence constraint annotations in various contexts. Specifically in the context of annotating the SARS-CoV-2 genome, in this manuscript we previously showed that our track based on state depletion of mutations showed stronger agreement with experimentally measured mutational effect on RBD than PhastCons and PhyloP scores.

In the revised manuscript, we now directly compare the information in ConsHMM annotations to that in PhastCons and PhyloP annotations for capturing annotations of genes and regions of interest within genes (**Supplementary Fig. 2**). We observe that in most cases the conservation states have greater precision at the same recall level than PhastCons and PhyloP annotations. This indicates that ConsHMM captures information in the alignment that is biologically relevant and distinct from information captured by these phylogeny-based methods. We focused on comparing annotations from

PhastCons and PhyloP since as with ConsHMM they also provide annotation for every base in the genome based on information in the alignments. Both methods were provided with the same input alignments as ConsHMM along with phylogeny trees.

We also present for each conservation state PhastCons and PhyloP score distributions of its bases and fraction of its bases in PhastCons element (**Supplementary Fig. 3**), which shows that distinct conservation states can have similar levels of sequence constraint as identified by PhastCons and PhyloP. These figures are now referenced and discussed in a new section titled “Conservation state’s relationship to standard sequence constraint annotations” as follows:

“To establish that conservation states contain additional information relative to standard sequence constraint scores, we compare to constraint scores generated by PhastCons⁴ and PhyloP⁵ and binary constrained elements called by PhastCons using the same alignments provided to ConsHMM in their ability to predict genes and regions of interest (**Methods**). When predicting bases overlapping genes or regions of interest within them, in most cases at least one of the conservation states achieve substantially greater precision at the same recall levels than PhastCons and PhyloP annotations (**Supplementary Fig. 2**). This suggests that when compared to existing constraint annotations based on the same alignments, ConsHMM conservation states captures additional biologically relevant information. In general, many states have largely overlapping distributions of PhastCons and PhyloP scores and similar fractions of constrained bases (**Supplementary Fig. 3**).”

We also describe how these analyses were conducted in more detail in Methods.

Supplementary Figure 2. Precision-recall plots for predicting genes and regions of interest. Shown in each subplot is a precision-recall plot for predicting bases that overlap external genomic annotations using ConsHMM conservation states and sequence constraint annotations. Above each subplot is the target annotation, which is either a gene (**a-l**) or a region of interest defined by UniProt²⁰ (**m-u**). In each subplot, prediction based on ConsHMM conservation states for bases overlapping the target annotation is shown with circles. Prediction based on sequence constraint scores is shown with continuous lines. Prediction based on PhastCons element is shown with triangles. Circles, lines, and triangles are colored according to the bottom right legend. Y-axis varies from subplot to subplot because the target annotations have different genome coverage. In most cases, at least one of the ConsHMM states have substantially greater precision at the same recall level than other sequence constraint annotations, suggesting that it provides greater information for recovering annotated bases.

Supplementary Figure 3. Conservation states' relationship to PhastCons and PhyloP annotations.

a. Shown for each conservation state learned from the Sarbecovirus alignment (x-axis) is the distribution of PhastCons score learned from the same alignment (y-axis) in bases overlapping the state. Each distribution is represented by a boxplot with median (orange horizontal line), mean (green 'x'), Q1 and Q3 (box), and Q1–1.5 IQR and Q3+1.5 IQR (whisker), where Q1 and Q3 represent 25th and 75th percentiles, respectively, and IQR (interquartile range) represent the difference between them.

b. Similar to **a** except showing conservation states and PhastCons score learned from the vertebrate CoV alignment.

c-d. Similar to **a-b**, respectively, except showing PhyloP score instead of PhastCons score.

e. Shown for each conservation state learned from the Sarbecovirus alignment (x-axis) is the fraction of bases overlapping PhastCons elements based on the same alignment (y-axis). Indicated by the horizontal dashed line is the genome-wide coverage of the PhastCons element annotation. The exact coverage is reported below the line.

f. Similar to **e** except showing conservation states and PhastCons elements learned from the vertebrate CoV alignment.

2. While ConsHMM can indeed "capture distinct patterns in the input alignment data" (Introduction), the biological interpretation of these patterns is less clear. For instance, the first section of the Results (based on 44 Sarbecoviruses) presents interpretations for just 4 out of 30 states. Do the other 26 states have plausible biological interpretations? And are the four states that are discussed ones that would not be detected using phylogeny-based approaches? (By "interpretation" I mean insight, not descriptions such as those in Supplementary Table 1.)

We thank the reviewer for raising these points. In addition to the four states noted by the reviewer, we gained insights into other states as well. For example, we also find that state S18 is similar in its align and match probabilities to state S17 except it has slightly reduced but still high (0.97) matched probabilities for two most distal strains from SARS-CoV-2. State S18 uniquely exhibits strong enrichment for a region in RNA-dependent RNA polymerase (RdRp) that interacts with remdesivir. Moreover, we find that state S6 corresponds to bases that align to all Sarbecoviruses but match only bat CoV RaTG13 with high probabilities, which may be of interest in understanding the uniqueness of SARS-CoV-2 relative to other Sarbecoviruses. We now discuss these additional states in the first Result section as follows:

“Similarly, state S18 annotates bases with high align and match probabilities except it has slightly reduced probability of matching to two strains that are most distal from SARS-CoV-2 (SARS-related CoV strain BtKY72 and Bat CoV BM48-31/BGR/2008). Unlike state S17, state S18 is strongly enriched for a region in RNA-dependent RNA polymerase (RdRp) that is known to interact with the antiviral drug remdesivir (10 fold; $P < 0.0001$). State S6 annotates bases where all strains align to SARS-CoV-2 with high probability but only the strain closest to SARS-CoV-2, bat CoV RaTG13, matches SARS-CoV-2 with high probability, highlighting bases with alleles unique to SARS-CoV-2 and bat CoV RaTG13 with respect to other Sarbecoviruses. As expected, state S6 is enriched for the third codon position (2.2 fold; $P < 0.0001$) where derived alleles are less likely to alter the amino acid.”

We also note that in this section we focused on states that had notable align and match probabilities and enrichments for genes and regions of interest. In the following sections that focus on SARS-CoV-2 mutations, four more states from the Sarbecovirus model are described.

Regarding the reviewer's question on whether the four states discussed in the first Result section are detectable by phylogeny-based methods, we present relevant results in **Supplementary Figure 2**, which we discussed above in our response to comment #1. When recovering bases in genes or regions of interest, the four states often have greater precision than PhastCons and PhyloP annotations at the same recall level. For example, when predicting bases in ACE2 receptor binding motif, state S28 achieves a precision of 0.18 at a recall of 0.22 whereas PhastCons and PhyloP scores have a precision of 0.002 and 0.003, respectively, at similar recall levels. Similarly, when recovering bases in ORF10, state S29 achieves a precision of 0.28 at a recall of 0.92 whereas both PhastCons and PhyloP scores have a precision of 0.007 at the same recall level. Overall, this indicates that the four states capture information distinct from that captured in PhastCons and PhyloP annotations.

3. With phylogeny-based methods, the biological interpretation of a result is often quite clear because there is a straightforward model for differences as the product of mutation, selection, drift, and recombination. In other words, the metrics map onto well understood processes. With the HMM approach, it feels more like an interpreting a Rorschach blot. Given that the goal of the present study is to create a resource, the more the authors can do to guide interpretation the more effective their results will be. Experiments take a lot of money, effort and time, making a "blind" priority score from the browser track will much less compelling than a mental model based on a biological mechanism.

We thank the reviewer for raising this point. We believe that phylogeny-based approaches are also useful and complementary to ConsHMM. Our motivation for using ConsHMM is to systematically annotate each nucleotide of the genome based on patterns in sequence alignments that are not captured by standard phylogeny-based constraint methods like PhastCons and PhyloP that generate univariate scores or binary element calls of constraint. As noted above, bases with similar constraint scores can be annotated with distinct conservation states. Our analysis of bases specific to pathogenic human CoV is one example of how we made use of the more detailed information associated with ConsHMM state annotations of specific sets of strains aligning to and/or matching SARS-CoV-2.

We also recognize having scores can be useful for some applications such as prioritization of bases where mutations are more likely consequential. This previously motivated us to generate a track of significant depletion of mutations in states, which was defined jointly based on both ConsHMM models. We realize that aggregating information from two ConsHMM models can make it difficult to interpret the track since it lacks a direct connection to a single conservation state model. To address this, we now present two additional tracks, one track scoring mutation depletion in the Sarbecovirus model's states and the other scoring mutation depletion in the vertebrate CoV model's states. Each of the tracks can be interpreted in terms of the state assignments from the corresponding ConsHMM model. They are now described in section 'Genome-wide tracks based on state depletion of SARS-CoV-2 mutations' as follows in addition to being included in the analysis of mutational effects on RBD:

"Specifically, for each ConsHMM model, we generated a track that scores each genomic base by its state's statistically significant depletion or enrichment of nonsingleton mutations, reflecting the mutation occurrence patterns among bases that likely share a common evolutionary history."

None of this is to say HMM-based approaches should not be used or even that they are inferior to conventional methods that are phylogenetically-based. To the contrary, I would very much like to see this study published and, more generally, the potential of HMM-based approaches to be more thoroughly investigated.

We thank the reviewer for this comment. We agree that there are pros and cons to both phylogeny-based methods and HMM-based methods. We believe that they can complement each other and provide more insight overall.

Minor issues:

1. It seems odd to present new results and comment on them in the Introduction (third paragraph). Indeed, Figure 1 is referenced only in the Introduction and not in the Results section, even though it presents new results.

We thank the reviewer for pointing this out. Figure 1 is now referenced in the Results section instead of Introduction as follows:

“First, we annotated the SARS-CoV-2 genome with 30 conservation states learned from the Sarbecovirus sequence alignment, labeled as states S1 to S30 (**Figs. 1-2; Supplementary Table 1; Methods**).”

2. When discussing HMM states, the narrative is often nonspecific (e.g., page 2 "One state", "another state", etc.). When discussing specific states please refer to them by number so that readers can find them in the tables and figures.

We thank the reviewer for the comment. We now avoid expressions like “a state”, “one state”, and “another state” and explicitly refer to each state by its number.

3. It would be helpful to have a little more information about masking bases (page 11). Some of criteria that UCSC uses to flag "Problematic Sites" are biological processes with a direct impact on understanding function (homoplasy, hypermutability), so masking bases can lead to the loss of important information. How many bases were masked in the various analyses? The effort by the authors to re-insert bases that were flagged specifically due to homoplasy for one analysis is much appreciated. What fraction of the overall flagged bases was this and much of an impact did this have on the results? Providing this information will give at least some limited insight into the effects of masking and how it might bias results.

We thank the reviewer for these questions. We now provide this information in the revised Methods section ‘Masking bases’ as follows:

“As a result, we masked 228 bases, analyzing 29,675 out of 29,903 bases (99.2%). The one exception was when we computed state enrichment for homoplastic mutations from a prior study. For this analysis only, we masked all problematic positions except for those described as homoplastic or highly homoplastic. As a result we masked 175 bases instead of 228 bases, analyzing 29,728 bases (99.4%).”

4. The comparison of PhyloP and ConsHMM predictions of mutational effect (page 8) is much appreciated. While the results are suggestive, they are extremely limited in scope. In other words, only one aspect of function (expression) was tested for only one region of one protein (RBD). Please provide appropriate caveats when discussing these results.

We appreciate the reviewer’s comment. We now note the caveats in the last sentence of this section as follows:

“This suggests that our genome-wide tracks based on depletion of mutations could help prioritize bases to mutate when aiming to identify mutations with strong impact on the virus’s protein expression and binding affinity or potentially other functionalities, but we note that this analysis does not provide direct evidence for other parts of the genome or other phenotypes of the virus.”

5. Another point about mutational effect is that readers need to understand exactly what the effect is. The manuscript says "expression" but study that is referenced primarily investigated the effects on Spike-ACE2 binding. Please clarify so that readers understand exactly what mutational effects were used to compare PhyloP and ConSHMM predictions.

We apologize that this was not made clear. The referenced study⁴ measured changes in both expression and binding affinity of RBD caused by mutations. We previously compared our genome-wide track to mutational effect on expression. Now we compare the track to mutational effect on binding affinity as well and report the results in the main text and **Supplementary Figure 5**. To specify the mutational effects examined in this section, we now make a distinction between the two as follows:

“The study specifically measured changes in RBD expression and binding affinity due to each possible amino acid change within RBD, where a positive value denoted increased expression or affinity and a negative value denoted decreased expression or affinity.”

Reference

1. Li, X. *et al.* Emergence of SARS-CoV-2 through recombination and strong purifying selection. *Sci. Adv.* (2020). doi:10.1126/sciadv.abb9153
2. Arneson, A. & Ernst, J. Systematic discovery of conservation states for single-nucleotide annotation of the human genome. *Commun. Biol.* **2**, 248 (2019).
3. Consortium, T. U. UniProt: a worldwide hub of protein knowledge. *Nucleic Acids Res.* **47**, D506–D515 (2018).
4. Starr, T. N. *et al.* Deep Mutational Scanning of SARS-CoV-2 Receptor Binding Domain Reveals Constraints on Folding and ACE2 Binding. *Cell* **182**, 1295-1310.e20 (2020).

REVIEWERS' COMMENTS:

Reviewer #1 (Remarks to the Author):

The authors have successfully addressed all the issues I raised in my previous round of review.

Reviewer #2 (Remarks to the Author):

The authors have addressed all of my concerns. I appreciate their diligence and efforts. I'm looking forward to seeing this study published.